Towards a barrier height benchmark set for biologically relevant systems

Kromann Jimmy C. 1
Christensen Anders S. 2
Cui Qiang 2
Jensen Jan H. 1 jhjensen@chem.ku.dk
1 Department of Chemistry, University of Copenhagen , Copenhagen , Denmark
2 Department of Chemistry, University of Wisconsin-Madison , Madison, WI , United States
Paci Emanuele
Electronic publication date: 2016 May 3
Publication date: 2016
Volume: 4
Electronic Location ID: e1994
Received 2016 Feb 25; Accepted 2016 Apr 10
Copyright: ©2016 Kromann et al.
Copyright year: 2016
Copyright holder: Kromann et al.
License: This is an open access article distributed under the terms of the Creative Commons Attribution License, which permits unrestricted use, distribution, reproduction and adaptation in any medium and for any purpose provided that it is properly attributed. For attribution, the original author(s), title, publication source (PeerJ) and either DOI or URL of the article must be cited.
License URL: https://creativecommons.org/licenses/by/4.0/

Keywords: Enzyme mechanism, Semiempirical methods, Benchmarks

Funding: University of Copenhagen NIH R01-GM106443 JCK acknowledges support from the University of Copenhagen. The work at UW-Madison is supported by the NIH grant R01-GM106443. The funders had no role in study design, data collection and analysis, decision to publish, or preparation of the manuscript.

==============================
We have collected computed barrier heights and reaction energies (and associated model structures) for five enzymes from studies published by Himo and co-workers. Using this data, obtained at the B3LYP/6- 311+G(2d,2p)[LANL2DZ]//B3LYP/6-31G(d,p) level of theory, we then benchmark PM6, PM7, PM7-TS, and DFTB3 and discuss the influence of system size, bulk solvation, and geometry re-optimization on the error. The mean absolute differences (MADs) observed for these five enzyme model systems are similar to those observed for PM6 and PM7 for smaller systems (10–15 kcal/mol), while DFTB results in a MAD that is significantly lower (6 kcal/mol). The MADs for PMx and DFTB3 are each dominated by large errors for a single system and if the system is disregarded the MADs fall to 4–5 kcal/mol. Overall, results for the condensed phase are neither more or less accurate relative to B3LYP than those in the gas phase. With the exception of PM7-TS, the MAD for small and large structural models are very similar, with a maximum deviation of 3 kcal/mol for PM6. Geometry optimization with PM6 shows that for one system this method predicts a different mechanism compared to B3LYP/6-31G(d,p). For the remaining systems, geometry optimization of the large structural model increases the MAD relative to single points, by 2.5 and 1.8 kcal/mol for barriers and reaction energies. For the small structural model, the corresponding MADs decrease by 0.4 and 1.2 kcal/mol, respectively. However, despite these small changes, significant changes in the structures are observed for some systems, such as proton transfer and hydrogen bonding rearrangements. The paper represents the first step in the process of creating a benchmark set of barriers computed for systems that are relatively large and representative of enzymatic reactions, a considerable challenge for any one research group but possible through a concerted effort by the community. We end by outlining steps needed to expand and improve the data set and how other researchers can contribute to the process.

Introduction

Semiempirical electronic structure methods are increasingly parameterized and benchmarked against data obtained by DFT or wavefunction-based calculations rather than experimental data (Stewart, 2007; Dral et al., 2016a; Gaus, Goez & Elstner, 2013). Using calculated data has the advantage that it represents the precise value (usually the electronic energy) that is being parameterized, with little random noise and even coverage of chemical space, including molecules that are difficult to synthesize or perform measurements on. Carefully curated benchmark sets, such as GMTKN30 (Goerigk & Grimme, 2011), are therefore an invaluable resource to the scientific community and heavily used.

For example, Korth & Thiel (2011) used the GMTKN24-hcno dataset (21 subsets of the GMTKN24 data set (Goerigk & Grimme, 2010), an earlier version of GMTKN30) to show that modern semi-empirical methods are approaching the accuracy of PBE/TZVP and B3LYP/TZVP calculations. While this is encouraging, one concern is whether the results obtained for the small systems that make up these data sets are representative of those one would obtain for large systems. For example, Yilmazer & Korth (2013) performed a benchmark study of hundreds of protein-ligand complexes that included protein atoms within up to 10 Å from the ligand and showed, for example, that the mean absolute difference (MAD) between interaction energies computed using PM6-DH+ and BP86-D2/TZVP was 14 kcal/mol. In comparison, the MADs for the S22 interaction energy subset of GMTKN24 are <2 kcal/mol for both dispersion corrected PM6 and DFT/TZVP calculations (Korth & Thiel, 2011). One likely explanation is that the systems in the S22 subset are too small to exhibit many-body polarization contributions to the binding energy that semi-empirical methods fail to capture. Another, or additional, possibility is that the S22 subset does not include ionic groups, which are quite common in proteins and ligands. Indeed Christensen, Elstner & Cui (2015) have recently assembled a salt-bridge benchmark set for which dispersion corrected PM6 and DFTB3 results in RMSD values of 4–6 kcal/mol. For DFTB3 the error can be reduced to 2–3 kcal/mol using chemical potential equilization.

The Yilmazer & Korth (2013) study raises a similar question about whether benchmark results for semiempirical barrier height-predictions on small systems, such as the BH76 and BHPERI subsets of GMTK24/30, are transferable to barrier height predictions for enzymes. The first step towards answering this question is to create a benchmark set of barriers computed for systems that are relatively large and representative of enzymatic reactions. This is a considerable challenge because, unlike for ligand-protein complexes, there is no large database of corresponding transition state (TS) structures (or even substrate-enzyme structures) to start from. Thus, TS structures must be computed which is time-consuming and hard to automate. There are a significant number of such structures in the literature but many are not computed at a high enough level of theory to serve as benchmarks. Furthermore, TS structures are known to depend significantly on the level of theory used and it is therefore important that the benchmark set is computed using identical or very similar levels of theory. Creating such a benchmark set is thus a considerable challenge for any one research group but can be addressed by a concerted effort by the community. This paper represents the first step in this process.

We have collected barrier heights and reaction energies (and associated structures) for five enzymes from studies published by Himo and co-workers (Chen, Fang & Himo, 2007; Georgieva & Himo, 2010; Hopmann & Himo, 2008; Liao, Yu & Himo, 2011; Sevastik & Himo, 2007), on a GitHub repository (github.com/jensengroup/db-enzymes). Using this data, obtained at the same level of theory, we then benchmark PM6, PM7, PM7-TS, and DFTB3 and discuss the influence of system size, bulk solvation, and geometry re-optimization on the error. We end by outlining steps needed to expand and improve the data set and how other researchers can contribute to the process.

Computational Methodology

Five systems are investigated: L-aspartate α-decarboxylase (AspDC), 4-oxalocrotonate tautomerase (4-OT), phosphotriesterase (PTE), histone lysine methyltransferase (HKMT), and haloalcohol dehalogenase (HheC). The reaction mechanisms that are investigated are shown schematically in Fig. 1. The B3LYP/6-311+G(2d,2p)//B3LYP/6-31G(d,p) (LANL2DZ is used for Zn in PTE) barrier heights and reaction energies are taken from the literature: AspDC (Liao, Yu & Himo, 2011), 4-OT (Sevastik & Himo, 2007), PTE (Chen, Fang & Himo, 2007), HKMT (Georgieva & Himo, 2010), and HhecC (Hopmann & Himo, 2008) and the corresponding atomic coordinates are taken from the supplementary information or supplied by Fahmi Himo. 4-OT and PTE have two-step mechanisms resulting in two barrier heights and reaction energies. All energies are taken relative to the reactant state which results in a negative barrier for the second step in the 4-OT mechanism (4-OT-2 in Table 1). The largest model system for each study is used unless noted otherwise and PCM results are for a dielectric constant of 80.

Table 1 Barrier heights and reaction energies calculated with a list of semi-empirical methods and compared to B3LYP/6-311+G(2d,2p)[LANL2DZ]//B3LYP/6-31G(d,p) values taken from the literature.

For barriers “−1” and “−2” refer to the first and second transition state in the mechanism, while for reaction energies they refer to the intermediate and product, respectively. MAD* values are computed without PTE. All values are in kcal/mol.

	AspDC	4-OT-1	4-OT-2	PTE-1	PTE-2	HKMT	HheC	MAD	MAD*	
Gas phase barrier heights	
B3LYP	13.5	6.9	−1.6	11.7	13.3	18.9	18.2		
PM6	11.9	1.4	−1.5	−18.1	−19.4	27.8	28.0	12.6	5.2	
PM7	5.6	8.1	5.8	−18.4	−27.3	24.0	31.7	15.1	7.0	
PM7-TS	23.1	−10.5	−18.5	−20.4	−18.5	30.5	34.4	19.4	14.3	
DFTB3	−4.9	9.2	3.3	18.7	14.1	18.3	24.6	5.8	6.5	
Gas phase reaction energies	
B3LYP	9.0	−5.7	−2.9	11.9	−4.6	−9.2	5.5	
PM6	2.0	−8.4	−1.7	−24.2	−14.2	−10.0	13.6	9.4	4.0	
PM7	−2.3	−2.3	2.7	−30.1	−26.5	−11.8	18.8	14.3	7.2	
DFTB3	−8.9	1.7	−3.0	21.7	−4.9	−7.6	12.7	6.3	6.8	
Solution phase barrier heights	
B3LYP	13.3	6.7	−0.8	10.5	11.1	19.1	17.0		
PM6	14.9	0.2	−1.3	−18.8	−21.3	27.7	25.1	12.4	5.1	
PM7	8.1	6.7	6.0	−20.0	−32.2	24.2	29.3	14.7	5.9	
PM7-TS	24.8	−13.3	−19.2	−19.9	−18.9	29.3	34.0	19.6	15.4	
Solution phase reaction energies	
B3LYP	10.0	−5.4	−2.6	8.9	−15.3	−12.4	4.2		
PM6	6.0	−9.6	−2.0	−27.0	−26.3	−12.6	10.3	8.9	3.0	
PM7	1.2	−3.3	2.2	−34.9	−40.7	−13.7	16.0	14.0	5.7	

Figure 1 Schematic representations of the reactions mechanisms for the five enzymes studied.

In the case of PTE Lys169 is carboxylated and two histidine ligands to each Zn ion are omitted for clarity.

The PM6 (Stewart, 2007), PM7 and PM7-TS (Stewart, 2012) single point calculations are performed using MOPAC2012 while the DFTB3 (Gaus, Cui & Elstner, 2011) single point calculations are performed using DFTB+ version 1.2.2 (Aradi, Hourahine & Frauenheim, 2007) and version 3ob-3-1 of the 3OB parameter set (Gaus, Goez & Elstner, 2013a; Gaus et al., 2014; Lu et al., 2015; Kubillus et al., 2015). PM7-TS calculations are only performed for barrier heights. The PMx/COSMO (Klamt & Schüürmann, 1993) are performed using a dielectric constant of 80. The PM6 geometry optimizations are done using Gaussian09 (Frisch et al., 2014) and, following Himo and co-workers, the position of some atoms were constrained to their crystallographic positions to mimic the constraints due to the protein atoms that have been removed. We constrain the same atoms as in the studies from which the coordinates are taken and refer the reader to these studies for an explanation of the choice of atoms. Figures 2–6 are made with Avogadro (Hanwell et al., 2012).

The B3LYP results include zero-point energy (ZPE) corrections and are therefore directly comparable to the relative enthalpy values predicted by PM6 and PM7. However, ZPE corrections are not included for the DFTB3 calculations and we note that the ZPE can contribute to the difference observed between the DFTB3 and B3LYP results. In our experience, ZPE corrections for the kind reactions considered here are typically no more than 2–3 kcal/mol for reactions involving hydrogen and less for other reactions. For example, exploratory calculations using the small structural models of 4-OTA and HheC showed that the ZPE contributed at most 2.4 kcal/mol to the energetics.

Results and Discussion

Gas phase

Table 1 lists barrier heights and reaction energies computed using PM6, PM7, PM7-TS, and DFTB3 single point energies on B3LYP/6-31G(d,p) geometries. For barrier heights the mean absolute differences (MADs) are 13, 15, 20, and 6 kcal/mol for PM6, PM7, PM7-TS and DFTB3. The 13 kcal/mol MAD for PM6 is comparable to the 10–15 kcal/mol MADs computed for PM6 by Korth & Thiel (2011) and Dral et al. (2016b) for various small molecule benchmark sets for barrier heights. For PM6 the accuracy is best for AspDC and 4-OT, for which the models only consist of atoms in the first and second row of the periodic table. For PM6 and PM7 the MADs are dominated by the PTE system (the only system containing a transition metal, Zn) where the errors range from 30 to 41 kcal/mol. Removing these two entries reduces the MADs to 5 and 7 kcal/mol, respectively for PM6 and PM7, which is 2–3 times lower than the MADs computed for PM6 by Korth & Thiel (2011). As we will show below, PM6 does not predict the same mechanism for PTE as B3LYP/6-31G(d,p). For DFTB3 the MAD is dominated by AspDC with an errors of 18 kcal/mol, while the first and second barrier for PTE is reproduced reasonably and very well, respectively. If the AspDC system is neglected the MADs for barrier heights decreases to 3.7 kcal/mol. The larger error observed for DFTB3 for AspDC is consistent with the observation that carbon dioxide remains to be one of the problematic cases for the 3OB parameterization with a large error in the atomization energy. Compared to the MIO parameterization (Elstner et al., 1998), 3OB significantly reduces the errors in atomization energy from a MAD of ∼47 kcal/mol to ∼5 kcal/mol for the Modified G2/97 CHNO Test Set; there are a few outliers, and carbon dioxide is one of those cases and has an error of 16.8 kcal/mol in atomization energy (Gaus, Goez & Elstner, 2013).

The errors computed for reaction energies have MADs of 9, 14, and 6 kcal/mol for PM6, PM7, and DFTB3. The lower MAD for PM6 compared to barrier heights is primarily due to the fact that the 10 error in the second step of the PTE mechanism (i.e., the difference between the product and the reactant) is considerably smaller than the 33 kcal/mol error in the corresponding barrier height. In all cases there is generally a correlation between errors in the reaction energies and errors in the corresponding barrier heights. Just as for the barriers, the MAD is reduced significantly for PM6 and PM7, to 4.0 and 7.2 kcal/mol, respectively, if PTE is disregarded. Similarly, the MAD for DFTB3 is reduced to 4.4 kcal/mol if AspDC is disregarded.

In summary, the MADs observed for these five enzyme model systems are similar to those observed for PM6 and PM7 for smaller systems (10–15 kcal/mol), while DFTB results in a MAD that is significantly lower (6 kcal/mol). The MADs for PMx and DFTB3 are dominated by large errors for one system (PTE and AspDC, respectively) and if the system is disregarded the MADs fall to 4–5 kcal/mol.

Effect of solvation

The inclusion of bulk solvation effects leads to very modest (≤0.5 kcal/mol) decreases in the MADs (Table 1). Some errors decrease, by as much as 3.0 kcal/mol for the reaction energy of AspDC, while others increase, by as much as 3.5 kcal/mol for the second reaction energy of PTE. This is in part due to the fact that the effect of bulk solvation on the B3LYP results is at most 4 kcal/mol for all methods, including B3LYP. The one exception is the second step in the PTE mechanism where solvation increases decreases the reaction energy by 10.7, 12.1, and 14.2 kcal/mol at the B3LYP, PM6, and PM7 level of theory. Thus, overall, results for the condensed phase are neither more or less accurate than those in the gas phase.

Effect of system size

In four of the five systems, Himo and co-workers computed barrier heights and reaction energies for between two and five systems of different size. The smallest system typically contain only the chemical entities directly involved in the chemical reaction. The next larger system includes groups that form hydrogen bonds to, or have steric interactions with, atoms in the smallest system, etc. We refer the reader to the original studies for an explanation of these systems. In Table 2 the columns marked “Model 0” lists the barrier heights and reaction energies for the smallest system, while the remaining “Model” columns lists the change in energetics on going to the next larger system. For example, at the B3LYP level the barrier height for AspDC is 0.5 kcal/mol larger when computed using Model 2 compared to Model 1. In the case of 4-OT “Model 0-1” and “Model 0-2” refer to the first and second barrier height computed using Model 0 in the case of barrier heights and, in the case of reaction energies, the energy of the intermediate and product relative to the reactant. The last column in Table 2 list the MAD of the energy changes (i.e., excluding Model 0) relative to B3LYP.

Table 2 Barrier heights and reaction energies calculated with a list of semi-empirical methods and compared to B3LYP/6-311+G(2d,2p)[LANL2DZ]//B3LYP/6-31G(d,p) values taken from the literature.

The column labeled “Model 0” is the energetics computed using the smallest structural model, while the remaining columns represent the change on going to the next-largest model. For ApsDC 4.1 and 4.2 refer to two structural models of roughly equal size and the change on going to model 5 is computed relative to Model 4.2. For barriers of 4-TA “−1” and “−2” refer to the first and second transition state in the mechanism, while for reaction energies they refer to the intermediate and product, respectively. All values are in kcal/mol.

	Model 0	Model 1	Model 2	Model 3	Model 4.1	Model 4.2	Model 5	MAD	
AspDC barriers		
B3LYP	0.1	8.2	0.5	0.2	4.9	4.0	0.5		
PM6	−1.5	5.2	1.9	1.1	2.0	2.5	2.7	2.0	
PM7	−4.5	4.1	3.2	−1.2	8.4	5.7	−1.7	2.6	
PM7-TS	4.5	17.6	3.4	−2.6	6.0	−2.9	3.1	4.3	
DFTB3	−6.8	−4.8	4.7	1.4	−1.3	1.6	−1.0	4.7	
AspDC reaction energies		
B3LYP	−9.5	9.8	−0.9	1.4	9.1	3.4	4.8		
PM6	−15.5	9.2	−0.5	1.6	6.4	1.8	5.4	1.0	
PM7	−15.9	10.8	2.1	−3.8	11.7	3.2	1.3	2.6	
DFTB3	−18.8	1.0	0.3	5.0	2.6	1.2	2.4	4.1	
	Model 0-1	Model 0-2	Model 1-1	Model 1-2					
4-OT barriers		
B3LYP	12.8	7.0	−5.9	−8.6					
PM6	5.4	0.9	−4.0	−2.4				4.1	
PM7	10.4	8.6	−2.3	−2.8				4.7	
PM7-TS	−16.8	−19.9	6.3	1.4				11.1	
DFTB3	18.3	12.6	−9.1	−9.3				1.9	
4-OT reaction energies		
B3LYP	9.8	−3.7	−15.5	0.8					
PM6	1.3	−3.6	−9.7	1.9				3.5	
PM7	5.4	−3.1	−7.7	5.8				6.4	
DFTB3	17.1	−5.5	−15.3	2.5				0.9	
	Model 0	Model 1	Model 2	Model 3					
HKMT barriers		
B3LYP	18.8	2.9	−6.3	3.5					
PM6	29.2	1.8	−5.2	2.0				1.2	
PM7	28.8	0.9	−7.2	1.5				1.6	
PM7-TS	39.6	−0.4	−10.3	1.6				3.1	
DFTB3	16.3	2.7	−4.7	4.0				0.8	
HKMT reaction energies		
B3LYP	−2.9	3.4	−17.2	7.5					
PM6	−5.5	5.5	−14.1	4.1				2.9	
PM7	−2.2	4.5	−17.3	3.2				1.8	
DFTB3	−1.2	5.5	−15.2	3.3				2.7	
	Model 0	Model 1	Model 2						
HheC barriers		
B3LYP	23.0	−5.1	0.3						
PM6	37.4	−6.5	−2.9					2.3	
PM7	42.8	−8.2	−2.9					3.2	
PM7-TS	49.6	−6.2	−9.0					5.2	
DFTB3	30.6	−7.3	1.3					1.6	
HheC reaction energies		
B3LYP	17.5	−3.4	−8.6						
PM6	30.8	−3.7	−13.5					2.6	
PM7	34.3	−6.6	−8.9					1.8	
DFTB3	24.8	−5.4	−6.7					1.9	

The data in Table 2 is summarized in Table 3. Columns two and three list the MAD relative to B3LYP for barrier heights computed using Model 0 and the largest model (excluding PTE), while columns four and five lists the corresponding values for reaction energies. The last two columns list the average MAD for the changes in barrier heights and reaction energies, respectively, due to increasing system size.

Table 3 “MAD TS Small” refers to the MAD from B3LYP in barrier heights computed using the small structural models listed in Table 2 and similarly for the reaction energies (“MAD RxnE Small”).

The values labeled “Big” are the corresponding MADs computed for the large systems taken from Table 1. The columns marked Δ are the MADs for the changes in barrier heights and reaction energies listed in Table 2. All values are in kcal/mol.

	MAD TS Small	MAD TS Big	MAD RxnE Small	MAD Rxn Big	ΔBarrier	ΔRxnE	
PM6	8.0	5.2	6.1	4.0	2.4	2.5	
PM7	7.7	7.0	5.8	7.2	3.0	3.1	
PM7-TS	21.7	14.3			5.9		
DFTB3	5.6	6.5	5.5	6.8	2.3	2.4	

With the exception of PM7-TS, the MAD for the small and large systems are very similar, with a maximum deviation of 3 kcal/mol for PM6. This indicates that the error observed in Table 1 stem primarily from the part of the system where bonds are being broken and formed. This is corroborated by the fact that the MAD for the change in barrier heights and reaction energies are all ≤3 kcal/mol (again with the exception of PM7-TS). It is not at all clear why the errors in barrier heights computed using PM7-TS differ significantly more from B3LYP for small structural models, compared to large.

Effect of geometry optimization

Table 4 compares the barrier heights and reaction energies computed using B3LYP and PM6 single points (from Table 1) to the corresponding values computed using PM6 optimized geometries using the largest and smallest structural models. The PTE system is excluded for reasons described below. The data for the large structural models indicate that with the exception of the 4-OT system the effect of optimization on the energetics is relatively minor (<4 kcal/mol) and does not necessarily improve the agreement with B3LYP.

Table 4 Gas phase barrier heights and reaction energies computed using B3LYP/6-311+G(2d,2p)// B3LYP/6-31G(d,p), PM6//B3LYP/6-31G(d,p), and PM6//PM6 (“PM6 opt”) using the largest and smallest structural models.

	AspDC	4-OT-1	4-OT-2	HKMT	HheC	MAD	
Large structural models	
Barrier heights	
B3LYP	13.5	6.9	−1.6	18.9	18.2		
PM6	11.9	1.4	−1.5	27.8	28.0	5.2	
PM6 opt	15.8	7.3	17.3	27.5	26.4	7.7	
Reaction energies	
B3LYP	9.0	−5.7	−2.9	−9.2	5.5		
PM6	2.0	−8.4	−1.7	−10.0	13.6	4.0	
PM6 opt	3.5	5.6	−3.0	−12.7	14.2	5.8	
Small structural models	
Barrier heights	
B3LYP	0.1	12.8	7.0	18.8	23.0		
PM6	−1.5	5.4	0.9	29.2	37.4	8.0	
PM6 opt	1.6	3.7	−1.1	32.6	28.8	7.6	
Reaction energies	
B3LYP	−9.5	9.8	−3.7	−2.9	17.5		
PM6	−15.5	1.3	−3.6	−5.5	30.8	6.1	
PM6 opt	−10.7	−4.6	−3.2	5.4	15.6	5.3	

In the case of 4-OT the agreement with B3LYP is improved considerably for the first barrier, reducing the error from 5.5 to 0.4 kcal/mol, while the agreement is worsened considerably for the second barrier (error increased from 0.1 to 18.9 kcal/mol) and the intermediate (error increased from 2.7 to 11.3 kcal/mol). Corresponding calculations using the smaller structural model of 4-OTA (Sevastik & Himo, 2007) leads to <2 kcal/mol changes in the barrier heights due to geometry optimization, with the exception of the intermediate, whose stability is increased by 5.9 kcal/mol due to a proton transfer from Arg11 to the substrate (Fig. 2). Thus, the change in energetics upon geometry optimization observed for 4-OT is most likely due to different interactions with ligands not immediately adjacent to the substrate.

Figure 2 PM6 optimized small structural model of the intermediate in the 4-OT reaction mechanism.

PM6 optimization leads to proton transfer from Arg11 (on the right) to the neighboring carboxyl group on the substrate.

Using the small structural models, the effect of optimization on the PM6 barrier heights is also relatively modest (<3 kcal/mol), with the exception of HheC, where the barrier drops by 8.6 kcal/mol. The drop in barrier height is likely due to a shift in position of Arg149 upon PM6 optimization so that it is now hydrogen bonded to the Ser132 oxygen and the oxygen on the substrate, rather than the oxygen of Tyr145 as in the B3LYP optimized structure (Fig. 3).

Figure 3 B3LYP/6-31G(d,p) (A) and PM6 (B) optimized small structural model of the transition state in the HheC reaction mechanism.

For reaction energies substantial decreases of 10.9 and 15.2 kcal/mol are observed for HKMT and HheC, respectively. In the case of HheC the change is due to a rather large structural rearrangement in which a proton from Arg149 is transferred to the Cl− (Fig. 4) while for HKMT the change appears to be due to a rather subtle change in the interaction between the S atom and the methyl group in the methylamine (Fig. 5).

Figure 4 B3LYP/6-31G(d,p) (A) and PM6 (B) optimized small structural model of the product in the HheC reaction mechanism.

Figure 5 B3LYP/6-31G(d,p) (A) and PM6 (B) optimized small structural model of the product in the HKMT reaction mechanism.

In the case of PTE the bonding in the reactant completely changes upon geometry optimization using PM6 (Figs. 6A and 6B). The Zn-phosphate bond is broken and a Zn–Zn bond is formed instead. Furthermore, a minimum is found on the PM6 potential energy surface (Fig. 6D) that is very similar to the second transition state found with B3LYP (Fig. 6C) except that the proton on the Zn-bridging OH group has not been transferred to Asp301 and the P–O bond to the nitrophenyl group is shorter by 0.40 Å in the PM6 optimized structure. Thus, taking B3LYP/6-31G(d,p) as the standard, PM6 fails to predict the correct mechanism for PTE, which also explains the very large errors observed for the PM6 single point calculations in Table 1. The PTE mechanism has been studied using AM1 (Wong & Gao, 2007), PM3 (Zhang et al., 2009), and AM1/d-based QM/MM methods (López-Canut et al., 2012) and that these studies have suggested other reaction mechanisms. We emphasize that we have not explored the mechanism of PTE using PM6 but simply compared the structures resulting from the PM6 geometry optimizations initiated from the B3LYP/6-31G(d,p) geometries obtained by Chen, Fang & Himo (2007).

Figure 6 B3LYP/6-31G(d,p) and PM6 optimized structure of the reactant (A) and (B), respectively in the PTE mechanism. B3LYP/6-31G(d,p) structure of the second transition state (C) and PM6 optimized structure of a minimum that is very similar to this transition state (D).

In summary, geometry optimization with PM6 shows that for PTE this method predicts a different mechanism compared to B3LYP/6-31G(d,p), consistent with the fact that PM6//B3LYP/6-31G(d,p) energetics differ very significantly from the reference values as discussed above. For the remaining systems, geometry optimization of the large structural model increases the MAD relative to single points, by 2.5 and 1.8 kcal/mol for barriers and reaction energies. For the small structural model, the corresponding MADs decrease by 0.4 and 1.2 kcal/mol, respectively. However, despite these small changes significant changes in the structure, such as proton transfer and hydrogen bonding rearrangements is observed for some systems.

Outlook

While our study identifies cases where semiempirical methods give results that differ significantly from the DFT and may require further attention, it is clear that five systems is not sufficient for a general and statistically significant assessment of the accuracy of a computational method. We plan to add more systems from the literature to the data set and invite other researchers to do the same. This can easily be done by making a free account on github.com and contributing to the project at github.com/jensengroup/db-enzymes. One simply creates a folder containing the coordinates in xyz format, barrier heights and reaction energies as well as a list of constrained atoms in csv files, and a README text file detailing the level of theory, commits the changes and opens a fork request for the repository, as is standard practice on Github.

While B3LYP/6-311+G(2d,2p)[LANL2DZ]//B3LYP/6-31G(d,p) may be an adequate level of theory to identify deficiencies in semiempirical methods it is unlikely to be accurate enough to parameterize against. In the case of intermolecular interactions the “gold standard” is CCSD(T)/CBS//MP2/TZVP computed using extrapolation (Jurecka et al., 2006; Řezáč & Hobza, 2013). This level of theory may be impractical for these size systems for the foreseeable future, but could be approximated by extrapolating from smaller systems using an ONIOM-like approach (Chung et al., 2015) or approaches like DLPNO-CCSD(T) (Liakos et al., 2015). At a minimum the DFT calculations (including the geometry optimizations) should be repeated with dispersion corrections and if a double-zeta basis set is used in the geometry optimization then a BSSE-correction should be used as well (Grimme, 2011; Goerigk & Reimers, 2013; Grimme et al., 2015; Karton & Goerigk, 2015). Again we encourage researchers interested in developing or testing such methods to use the coordinates in the data set and deposit the barriers and reaction energies.

We thank Fahmi Himo for providing coordinates for the HKMT and HheC structural models and for providing useful comments on the manuscript. We thank Christof Jaeger, Stefan Grimme, and Pedro J. Silva for valuable feedback on the manuscript.

Additional Information and Declarations

Competing Interests

Author Contributions

Data Availability

The authors declare there are no competing interests.

Jimmy C. Kromann and Anders S. Christensen performed the experiments, analyzed the data, contributed reagents/materials/analysis tools, prepared figures and/or tables, reviewed drafts of the paper.

Qiang Cui analyzed the data, reviewed drafts of the paper.

Jan H. Jensen conceived and designed the experiments, performed the experiments, analyzed the data, wrote the paper, prepared figures and/or tables, reviewed drafts of the paper.

The following information was supplied regarding data availability:

All input and output files are available as supplementary information on Figshare: https://dx.doi.org/10.6084/m9.figshare.2157379.

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
