# Peer review of "Towards a barrier height benchmark set for biologically relevant systems"

_PeerJ, doi:10.7717/peerj.1994_

## Round 0.1 · original submission · Minor Revisions

The manuscript has been thoroughly assessed by three reviewers. While all the reviewers recommend publication after minor revisions, I hope you will address thoroughly the reviewers' extensive criticisms in your resubmission.

·

Basic reporting

No Comments

Experimental design

No Comments

Validity of the findings

No COmments

Additional comments

The present papers is a first step towards the generation of a reaction database for biological systems. To do so the authors provide with highly accurate DFT and ab-initio structural and thermodynamic data for a set of five enzyme reactions. The results are compared with various PM# type semi-empirical methods and DFTB3. The work sets up an standard in terms of semi-empirical method validation, that hopefully, it will cristalize in a joint effort with the rest of the theoretical chemist community to create a database of biologically relevant reactions. In this sense, I find this paper with a high potential and should be publised.

Although the set of reactions is small as to draw general conclusions, I understand that the goal of the authors is to set up an standard so that other theoretical chemistry groups worldwide will contribute to this joint effort. I think that this is a very good and necessary idea. I would suggest that it would be useful in the github.com/jensengroup/db-enzymes dataset that the authors would consider to provide with standard inputs for the programs that should be considered as standard for other theoreticians as well. In this way, the desirable growth of this dataset by contributions from other groups will not jeopardize its necessary coherency.

Reviewer 2 ·

Basic reporting

The authors present a compilation of model systems for five, real-life enzymatic reactions based on previously published structures, reaction barrier heights and reaction energies that had been obtained with the B3LYP density functional theory (DFT) approximation. These DFT structures and numbers are used as a reference to examine various cost-efficient semi-empirical methods. The authors consider both geometry optimisations, as well as single-point energy calculations. Furthermore, they also comment on the effect of a solvation model on the results.

The main motivation of this study is clearly outlined and it is important to note that the authors point out that their study is only the first step in a long-term undertaking that aims at having a diverse set with more accurate reference energies. The language is clear and professional. Raw data is supplied through a link to figshare and can be easily accessed and examined by the readers of this paper. With regards to cited literature and the figures, I have the following suggestions to make:

a) I think the cited literature should be more comprehensive and also take into consideration recent QM studies on peptides and proteins that allow a better understanding of the quality of the herein used DFT reference values; I will elaborate on this point more in detail under “validity of the findings”. I also noticed that some cited articles lack a volume and page number; I therefore recommend that the authors carefully check their list of references for any mistakes.

b) While the general quality of the figures is very good, I find Figures 3 - 6 hard to read. In these figure, the authors attempt to compare geometries for two levels of theory with each other. While one does note differences, the structures lack a common reference point that would allow the reader to better gauge whether these differences are due to structural changes or due to slightly different viewing angles. Would it be possible to improve these figures and to align the geometries with respect to one of the involved molecules?

Experimental design

I identified some issues that would make it hard for others to exactly reproduce the results:

a) page 2, line 78: The authors say that during the geometry optimisations, the position of some atoms were restrained. It is not clear which atoms these are and why that choice was made.

b) In section 3.3 (and in Tables 2 and 3) the authors refer to different models for each of the five tested reactions that all vary in system size. This is the first time in the manuscript that the authors refer to this and it is hard to follow how exactly these models differ from each other and what they actually look like. It therefore makes it hard to follow this section and I recommend a revision.

Validity of the findings

As mentioned above, I do understand what the authors are trying to achieve with their study. It is supposed to only serve as a starting point for the future development of an accurate benchmark set. Also, it is likely that the authors’ current computational capabilities are limited and that they therefore use previously published structures and energies. Unfortunately, the chosen level of theories for the geometry optimisations [B3LYP/6-31G(d,p)] and the singlepoint energy calculations [B3LYP/6-311+G(2d,2p)] are questionable for the studied systems and properties. I will first elaborate on this statement, starting with the quality of the geometries, before I will conclude with a recommendation on how the authors could address this problem without substantially delaying publication.

The structural stability of biomolecular systems - and of any sizeable molecule or molecular aggregate – is heavily influenced by important London-dispersion (van-der-Waals) effects, and it is a well-known fact among quantum chemists that B3LYP does not cover these effects. Moreover, small basis sets, such as 6-31G(d,p) induce the so-called basis-set superposition error (BSSE), which is an artificial overstabilisation of noncovalent interaction energies, including London dispersion and hydrogen bonds. BSSE exists both for inter- and intramolecular interactions and it has been demonstrated in the literature that it distorts molecular structures, see e.g. early works by van Mourik and co-workers on peptide conformers:
-van Mourik, T.; Karamertzanis, P. G.; Price, S. L., J. Phys. Chem. A 2006, 110, 8.
- Holroyd, L. F.; van Mourik, T., Chem. Phys. Lett. 2007, 442, 42.

Later, Goerigk and Reimers showed how error compensation between the lack of a proper dispersion treatment and BSSE can artificially generate structures of seemingly acceptable quality. However, it was also demonstrated that this error compensation cannot be controlled and that in fact using dispersion and BSSE corrections led to more accurate and reliable structures. This was first discussed for gas-phase structures of peptides, and later for the crystal structure of a protein fragment:
- Goerigk, L.; Reimers, J. R., J. Chem. Theory Comput. 2013, 9, 3240.
- Goerigk, L.; Collyer, C. A..; Reimers, J. R., J. Phys. Chem. B 2014, 118, 14612.

Efficient corrections for both London dispersion and BSSE exist that do not compromise the computational efficiency of B3LYP/6-31G(d,p) and they should also be mentioned (e.g. Grimme’s DFT-D3(BJ) and gCP corrections). This is particularly important, as this journal is directed at a readership that may not be familiar with these methodological developments and the dispersion and BSSE problems.

While BSSE is negligible for the 6-311+G(2d,2p) basis set, the singlepoint energies still suffer from the London-dispersion problem. The authors already mention the large GMTKN30 benchmark set in their manuscript. An additional study on GMTKN30 with nearly 50 methods has conclusively shown how dispersion can influence reaction energies and barrier heights by several kcal/mol (Goerigk, L.; Grimme, S.; Phys. Chem. Chem. Phys. 2011, 13, 6670.). A proper treatment of dispersion is therefore crucial. That same study has also shown that B3LYP is worse than the average of more than 20 hybrid density functionals for reaction energies. Consequently, using B3LYP/6-311+G(2d,2p) as a benchmark for the study of other methods may therefore distort the findings, and statistical values such as mean absolute deviations have to be interpreted with caution. In fact, it may well be that some of the tested semi-empirical methods perform much better than what the reported MADs may suggest. In this context, note that a recent study by Karton and Goerigk has demonstrated how the quality of the benchmark reference heavily influences the interpretation of low-level methods used to calculate barrier heights of pericyclic reactions (J. Comp. Chem. 2015, 36, 622). If one analyses the results of this study carefully, one can also see the importance of dispersion; for instance, it lowers the B3LYP barrier height of the Diels-Alder reaction between 1,3-butadiene and ethene by nearly 6 kcal/mol!

After this explanation, please let me emphasise that I am not against the way this study has been carried out. The studied reactions are highly interesting and the actual reaction barriers and energies provide useful insights for quantum chemists. Recalculating the reported numbers may therefore not be necessary at this stage and it can be postponed to a future study. Nevertheless, I am of the opinion that this manuscript can benefit from a discussion of the above points, particularly in the outlook section. Dispersion and BSSE should not be neglected and this manuscript is an ideal platform to convey this message to a non-expert readership that may not be aware of how the field of DFT approximations has changed over the past years. Such a discussion should also mention that also the tested semi-empirical methods lack from a proper description of noncovalent interaction energies and that others have combined them with DFT-D2 or DFT-D3-type corrections (DFTB3-D3, PM6-D3, etc. ) and with additional hydrogen-bond corrections for PM6 (see e.g. the works by Hobza and Korth). One or two paragraphs discussing the importance of these points are therefore sufficient.

Additional comments

page 3, line 104: Is the quotation mark at the beginning a mistake or does it introduce a literal quote from another paper? If the latter is the case, a second quotation mark at the end of the quote is missing.

Reviewer 3 ·

Basic reporting

I think this paper represents a nice effort of building a set for benchmarking barrier height in modelling biochemical systems. The authors provide themselves the benchmark for several semiempirical methods, using B3LYP as a reference point. Nevertheless, as the authors acknowledge, they provide a limited number of systems studied and a reference state (B3LYP) which could not be considered the state of the art. Thus, in my opinion the manuscript should be considered more as proposing the benchmark idea than a rigorous initial step towards it. In this regards I miss some section underlying how to expand the set, with some practical example. It could be even more interesting some efforts in making that part easier (semi-automatic) from standard QM or QM/MM calculations (although this part could be expanded in the future).

Experimental design

Since most reports are based on the comparison with B3LYP data (it seems to me that from other sources) I wander if the authors have reevaluated some of these numbers. Being single point evaluations it should not require excessive computational time.

Validity of the findings

To help the inspection of the results (and differences between B3LYP and semiempirical methods), I would like to see the distances (at least the most changing ones) being added in Figures 2-5.

Additional comments

minor comments:

.) (line 47): "...TS structures are known to dependent significantly"
.) (line 104) " ...kcal/mol. “The larger..." (a non closed ")
.) (line 116) "PM7 (to 4.0 and" (non closed parenthesis)

---

## Round 0.2 · accepted · Accept

Thank you for submitting a revised version and a rebuttal letter that addressed the reviewers' comments.